

# Identification of novel *BRCA1* large genomic rearrangements by a computational algorithm of amplicon-based Next-Generation Sequencing data

Arianna Nicolussi[1], Francesca Belardinilli[2], Valentina Silvestri[2], Yasaman Mahdavian[2], Virginia Valentini[2], Sonia D'Inzeo[3], Marialaura Petroni[4], Massimo Zani[2], Sergio Ferraro[2], Stefano Di Giulio[2], Francesca Fabretti[2], Beatrice Fratini[1], Angela Gradilone[2], Laura Ottini[2], Giuseppe Giannini[2,5], Anna Coppa[1] and Carlo Capalbo[2]

[1] Department of Experimental Medicine, University of Roma "La Sapienza", Roma, Italy
[2] Department of Molecular Medicine, University of Roma "La Sapienza", Roma, Italy
[3] U.O.C. Microbiology and Virology Laboratory, A.O. San Camillo Forlanini, Roma, Italy
[4] Istituto Italiano di Tecnologia, Center for Life Nano Science @ Sapienza, Roma, Italy
[5] Istituto Pasteur-Fondazione Cenci Bolognetti, Roma, Italy

Corresponding authors
Giuseppe Giannini,
giuseppe.giannini@uniroma1.it
Anna Coppa,
anna.coppa@uniroma1.it

## ABSTRACT

**Background**. Genetic testing for *BRCA1/2* germline mutations in hereditary breast/ovarian cancer patients requires screening for single nucleotide variants, small insertions/deletions and large genomic rearrangements (LGRs). These studies have long been run by Sanger sequencing and multiplex ligation-dependent probe amplification (MLPA). The recent introduction of next-generation sequencing (NGS) platforms dramatically improved the speed and the efficiency of DNA testing for nucleotide variants, while the possibility to correctly detect LGRs by this mean is still debated. The purpose of this study was to establish whether and to which extent the development of an analytical algorithm could help us translating NGS sequencing via an Ion Torrent PGM platform into a tool suitable to identify LGRs in hereditary breast-ovarian cancer patients.

**Methods**. We first used NGS data of a group of three patients (training set), previously screened in our laboratory by conventional methods, to develop an algorithm for the calculation of the dosage quotient (DQ) to be compared with the Ion Reporter (IR) analysis. Then, we tested the optimized pipeline with a consecutive cohort of 85 uncharacterized probands (validation set) also subjected to MLPA analysis. Characterization of the breakpoints of three novel *BRCA1* LGRs was obtained via long-range PCR and direct sequencing of the DNA products.

**Results**. In our cohort, the newly defined DQ-based algorithm detected 3/3 *BRCA1* LGRs, demonstrating 100% sensitivity and 100% negative predictive value (NPV) (95% CI [87.6–99.9]) compared to 2/3 cases detected by IR (66.7% sensitivity and 98.2% NPV (95% CI [85.6–99.9])). Interestingly, DQ and IR shared 12 positive results, but exons deletion calls matched only in five cases, two of which confirmed by MLPA. The

breakpoints of the 3 novel *BRCA1* deletions, involving exons 16–17, 21–22 and 20, have been characterized.

**Conclusions**. Our study defined a DQ-based algorithm to identify *BRCA1* LGRs using NGS data. Whether confirmed on larger data sets, this tool could guide the selection of samples to be subjected to MLPA analysis, leading to significant savings in time and money.

## BACKGROUND

Hereditary breast and ovarian cancer syndrome, caused by germline pathogenic mutations in the *BRCA1* (MIM#113705) or *BRCA2* (MIM#600185) genes, is characterized by an increased risk for breast, ovarian, pancreatic and other cancers (*Palma et al., 2006*). It has been recently estimated that the cumulative risks of breast cancer to age 80 years was 72% for *BRCA1* and 69% for *BRCA2* carriers (*Kuchenbaecker et al., 2017*). Differences in mutation type and site may at least partially impact on cancer risk definition (*Rebbeck et al., 2015*; *Coppa et al., 2018*; *Rebbeck et al., 2018*). *BRCA1* and *BRCA2* gene mutations are typically found in 25–30% of the breast cancer families subjected to genetic testing (*Giannini et al., 2006*; *Economopoulou, Dimitriadis & Psyrri, 2015*). The relatively low rate of success in finding relevant pathogenic mutations in this settings is likely due to the contribution of other moderate-to-high penetrance breast cancer susceptibility genes (i.e., *PALB2*, *ATM*, *CHK2*) (*Economopoulou, Dimitriadis & Psyrri, 2015*; *Coppa et al., 2018*), or to the influence of low penetrance and risk-modifying alleles (*Couch et al., 2012*; *Ottini et al., 2013*; *Kuchenbaecker et al., 2014*; *Peterlongo et al., 2015*), all of which needs to be taken into account for a more appropriate assessment of individual cancer risk. For quite some time, the use of classical qualitative PCR-based techniques incapable of detecting large genomic rearrangements (LGRs) also contributed to failures in the identification of *BRCA* mutation carriers. Interestingly, the prevalence of *BRCA1/BRCA2* LGRs varies greatly among different populations ranging from 0 to 27% of mutation positive families in Iranian/French, Canadian, Dutch, Spanish, German, French and South Africa populations (*Gad et al., 2002*; *Hogervorst et al., 2003*; *Hartmann et al., 2004*; *Pietschmann et al., 2005*; *Moisan et al., 2006*; *La Hoya de et al., 2006*; *Sluiter & Van Rensburg, 2011*). Relevant differences in the frequency of *BRCA1* LGRs have also been reported within the Italian population (*Montagna et al., 2003*; *Buffone et al., 2007*). In general, *BRCA2* LGRs are less frequent (*Woodward et al., 2005*; *Agata et al., 2005*; *Buffone et al., 2007*), probably due to the lower density of Alu sequences compared to *BRCA1*, which are involved in the genesis of LGRs (*Smith et al., 1996*). Multiplex ligation-dependent probe amplification (MLPA) is the most commonly used technique for the detection of large deletions/duplications in *BRCA1/2* genes.

The recent advances in sequencing technologies have increased the speed and efficiency of DNA testing and the emergence of benchtop next-generation sequencing (NGS) instruments are becoming the standard in molecular genetic diagnosis

(*Feliubadalo et al., 2013*; *Trujillano et al., 2015*). NGS is capable of sensitive detection of sequence variants, but may also be used for detection of LGRs by the evaluation of Copy Number Variations (CNVs) (*Tarabeux et al., 2014*; *Enyedi et al., 2016*; *Schenkel et al., 2016*; *Schmidt et al., 2017*). The CNVs assessment is mainly performed using the sequencing read depth (RD) assessment approach, whose assumption is that the RD signal is proportional to the number of copies of chromosomal segments present in that specimen (*Tan et al., 2014*). The ability to detect CNVs from NGS multigene panel largely, but not uniquely, depends on the library preparation, and target enrichment approaches based on hybridization and capture seem to have better performances compared to amplicon-based methods. In general, NGS data are not routinely used for CNVs detection in clinical settings for *BRCA* mutation screenings, due to concerns related to library preparation protocols, normalization procedures and employed software (*Feliubadalo et al., 2013*; *Wallace, 2016*). Recently, we adopted the NGS Ion AmpliSeq$^{TM}$ BRCA1 and BRCA2 Panel to perform routine *BRCA1/2* mutation screening on the Ion PGM platform (*Nicolussi et al., 2019*). Here, we aimed at establishing whether sequencing data generated by this approach could be processed by a computational algorithm to efficiently predict the presence of LGRs, based on the dosage quotient (DQ) calculation and the Ion Reporter (IR) analysis.

## METHODS

### Patients and DNA

Families putatively affected by hereditary breast/ovarian cancer syndrome were recruited at the Hereditary Tumors section of Policlinico Umberto I, University La Sapienza, between July 2015 and September 2017 and selected as previously described (*Capalbo et al., 2006b*; *Capalbo et al., 2006a*; *Coppa et al., 2014*). Comprehensive pre-test counseling was offered to all probands and their family members and informed consent was obtained. For each study participant, samples of blood or DNA from peripheral blood leukocytes were collected. DNA from blood samples was extracted and quantified as described by *Nicolussi et al. (2019)*. All investigations were approved by Ethics Committee of the University of Roma "La Sapienza" (Prot.: 88/18; RIF.CE:4903, 31-01-2018) and conducted according to the principles outlined in the declaration of Helsinki.

A retrospective group of 3 DNA samples, previously found positive for *BRCA1* LGRs by MLPA was used as a training set (TS). LGRs in the TS were as follows: sample BR59, *BRCA1* exon 23–24 deletion (c.5407-?*(1_?)del); sample BR328, *BRCA1* exon 18-19 deletion (c.5075-?_5193+?del)(*Buffone et al., 2007*) and sample BR409, *NBR2* exon1 and *BRCA1* exon 1-2 deletion (NBR2del EX1_BRCA1 delEX1-2) (*Coppa et al., 2018*) (Table 1).

For NGS-based LGR analysis, a consecutive group of 127 NGS/MPLA negative samples have been used to create a baseline and a prospective consecutive cohort of 85 uncharacterized probands, validation set (VS), was studied.

### Ion torrent PGM sequencing

The target regions in the *BRCA1* and *BRCA2* genes were amplified using the Ion AmpliSeq$^{TM}$ BRCA1 and BRCA2 Panel (Life Technologies) according to the manufacturers'

**Table 1 LGRs in TS and VS.**

|  | Sample Id | Genomic variant | Exon deletion | Ref |
|---|---|---|---|---|
| TS | BR59 | c.5407-?_*(1_?)del | exon 23-24 del | *Buffone et al. (2007)* |
|  | BR328 | c.5075-?_5193+?del | exon 18-19 del | *Buffone et al. (2007)* |
|  | BR409 | NBR2delEX1_BRCA1delEX1-2 | exon 1 NBR2 del<br>exon 1-2 BRCA1 del | *Coppa et al. (2018)* |
| VS | BR963 | NG_005905.2: g.163181_169408del6228 | exon 21-22 del | / |
|  | BR1154 | NG_005905.2: g.160396_164568del4173 | exon 20 del | / |
|  | BR1379 | NG_005905.2:g.145185_151339del6155 | exon 16-17 del | / |

procedures and processed as previously described (*Belardinilli et al., 2015*; *Nicolussi et al., 2019*).

## Sanger sequencing

All clinical samples were sequenced for the entire coding regions by Sanger sequencing, using an ABI PRISM DyeDeoxy Terminator Cycle Sequencing Kit and an ABI 3100 Genetic Analyzer (Applied Biosystems, Warrington, UK). Reference sequence for *BRCA1* was Genbank, NM_007294.3, and reference sequence for *BRCA2* was Genebank, NM_000059.3.

## MLPA analyses

MLPA methodology (*Schouten et al., 2002*) was performed, according to the manufacturer's instructions (MRC–Holland, Amsterdam, the Netherlands), to identify *BRCA1/2* genomic rearrangements. For the statistical analysis we transferred the size and the peak areas of each sample to an Excel file. The peak areas of the expected MLPA products were evaluated by comparison with a normal control and by cumulative comparison of all samples within the same experiment (*Buffone et al., 2007*; *Coppa et al., 2018*).

## NGS-based LGRs analysis

LGRs in *BRCA1* gene were studied by two distinct approaches: the manual calculation of the DQ and the IR platform. In the manual approach, DQ for each sample was calculated as follows: amplicon read count normalized on the *BRCA1* and *BRCA2* total reads/average of normalized amplicon read counts obtained from all samples. Specifically, we referred to DQA when amplicon counts were normalized vs. the coverage data of all samples run on the same single chip, and to DQB when amplicon counts were normalized vs. coverage data obtained from a baseline built from 127 LGRs negative samples. In addition, DQB has been alternatively obtained either considering together all amplicons of the Ion AmpliSeq[TM] BRCA1 and BRCA2 Panel (DQB1) or by separately considering the three different pools of amplicons (DQB2). DQ value higher than mean plus two standard deviations (SD) was considered indicative of a duplication; DQ value lower than mean minus 2 SD was considered indicative of a deletion. Particular attention has been also payed to reduction of multiple consecutive amplicons, even when they failed to trespass the above defined thresholds.

In the IR approach, we create a user-defined CNV detection workflow by a tunable Ion Reporter[TM] Software algorithm based on Hidden Markov Model (HMM), that utilize

normalized read coverage across amplicons to predict the copy number or ploidy (https://assets.thermofisher.com/TFS-Assets/LSG/brochures/CNV-Detection-by-Ion.pdf). The data coverage of 20 mutation-negative patients has been used as CNV baseline to analyze the samples of both TS and VS. We detected no *BRCA2* LGR in both the TS and VS. Thus, our analysis is necessarily limited to *BRCA1* LGRs.

### DNA breakpoint analysis

Newly discovered *BRCA1* large deletions were validated by characterization of the genomic breakpoints. Long-range PCR was performed according to the manufacturer's instructions using the kit Platinum Taq DNA polymerases High Fidelity (Thermo Fisher) with the primers sitting on closer undeleted exons as described in Table S2. PCR products were purified with ExoSAP-IT (USB Corp., Cleveland, USA) according to the manufacturer's instructions and sequenced using the ABI PRISM DyeDeoxy Terminator Cycle Sequencing Kit and an ABI 3100 Genetic Analyzer (Applied Biosystems, Warrington, UK). Reference sequences for *BRCA1* and *BRCA2* are in GenBank; NM_007294.3 and NM_000059.3, respectively.

### Statistical analysis

Validation metrics were defined as: Accuracy = (TP + TN)/(TP + FP + TN + FN); Sensitivity = TP/(TP + FN); Specificity = TN/(TN + FP); Negative Predictive Value = TN/(TN + FN), where TP = true positives, TN = true negatives, FP = false positives, FN = false negatives. The confidence intervals (CIs) were calculated by the method of *Wilson (1927)*.

## RESULTS

### NGS-dependent LGR analyses

To establish whether the data obtained by NGS via Ion AmpliSeq$^{TM}$ *BRCA1* and *BRCA2* Panel were suitable to identify copy number alterations in *BRCA1*, we used data from three samples (TS), already characterized in our laboratory for the presence of *BRCA1* LGRs by MLPA (Table 1). The sequencing data of the TS were analyzed by a locally devised algorithm for the calculation of the DQ and by our custom modified IR analysis, as described in materials and methods. The intrarun DQ calculation (DQA), which includes normalization based on the coverage data of the samples sequenced in the same chip, was always included to monitor the variability eventually due to different batches of reagents or to time-related variables. In general, however, we thought we could get improved resolution and reduced numbers of CNV false calls by normalizing the coverage data of all amplicons of each sample vs. those obtained from a reference set of 127 MLPA negative samples selected on the basis of their quality and uniformity of the coverage (DQB analysis). This baseline has been used to perform two DQB calculations, considering either all amplicons contained in the Ion AmpliSeq$^{TM}$ BRCA1 and BRCA2 Panel (DQB1) or dividing them into the three subsets identified by the amplification primer pools (DQB2).

As shown in Fig. 1A, the DQA plot of the TS samples revealed the presence of peaks below the thresholds, in samples BR328 and BR409 (corresponding to deletions of *BRCA1*

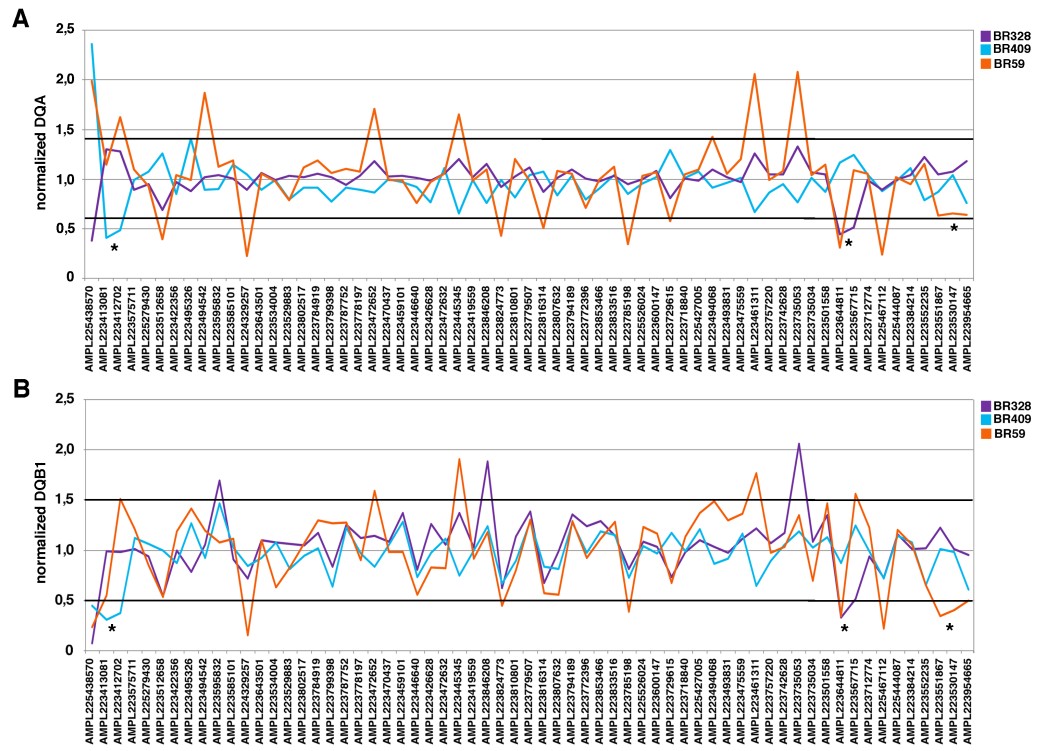

**Figure 1 DQ analyses for TS samples.** (A) For each sample, every peak represents the ratio of the amplicon read count normalized on *BRCA1/BRCA2* total reads and the average of normalized amplicon read counts from all samples on a single chip (DQA). (B) For each sample, every peak represents the ratio of the amplicon read count normalized on *BRCA1/BRCA2* total reads and the average of the coverage data of a baseline built from 127 LGRs negative samples (DQB1). The threshold = mean ± 2 SD. Value > mean ± 2 SD is indicative of a duplication; Value < mean ± 2 SD is indicative of a deletion. * indicated the amplicons included in the region involved in the rearrangement as confirmed by MLPA analysis.

exons 18-19 and 1-2, respectively, in agreement with MLPA results). The DNA quality of BR59 sample was rather low, as evidenced by the many peaks out of the threshold. Nevertheless, the DQB1 analysis evidenced values below the threshold for 3 consecutive amplicons (AMPL223551867, 223530147 and 223954665), identifying *BRCA1* exon 23-24 deletion (Fig. 1B), already discovered by MLPA analysis. Although they fail to trespass the threshold, the same consecutive amplicons showed strongly reduced values also at DQA evaluation (Fig. 1A). Hence, the careful examination of the two DQ calculations allowed us to identify all three *BRCA1* LGRs in the TS. Also, the analysis performed by IR software detected the presence of CNV (CNV = 1) in the proper regions in all three TS samples (Table 2). On this basis, we extended DQA, DQB and IR analysis to a group of 85 consecutive samples (VS) negative for *BRCA1/2* pathogenic variants at NGS analysis and compared it with MLPA results. Overall, DQA and DQB analysis resulted in detection of positive calls in 33/85 (39%) samples, while IR analysis detected CNVs in 29/85 (34%) (Table 3). Interestingly, DQ and IR evaluation only shared 12 positive results, with exon calls being not coincident in seven of them and with a rather precise, although imperfect, indication of the exons involved in the remaining five (Table S1). MLPA confirmed *BRCA1*

LGRs in 3/85 samples (Fig. 2): BR963 and BR1379, belonging to the small group of five DQ/IR double positive samples, and BR1154 resulted DQ positive-IR negative. Therefore, DQ calculation resulted 100% sensitive and displayed a 100% NPV (95% CI [87.6–99.9]) (Table 3) in our VS, values not reached by IR analysis, which failed in the identification of BR1154 (Table 2). Within DQ analysis, the correct calls were more clearly defined by the DQB2 calculation (Figs. 3A–3C). The appropriateness of the deletions calls of DQ, IR and MLPA evaluations were confirmed by the molecular characterization of the breakpoints, as described below.

## Characterization of LGRs

Identification of the breakpoints characterizing the LGRs is important for several reasons, including the possibility to develop diagnostic assays for segregation analyses in relatives. For different reasons DQ, IR and MLPA analyses are not able to provide such detailed molecular characterization of LGR. To define the breakpoints of the newly identified *BRCA1* LGRs, PCR amplification of genomic DNA from the three samples and direct sequencing were performed.

As shown in Fig. 4A, PCR amplification of genomic DNA from the BR963 patient resulted in an aberrant fragment of approximately 1,353 bp, whose direct sequencing confirmed loss of *BRCA1* exons 21 and 22, possibly originating from an erroneous homologous recombination process between an AluSq2 (Alu family, SINE class; chr17:41206762-41207066) and an AluSz (Alu family, SINE class; chr17:41200521-41200834) motifs. The rearrangement involved a perfectly repeated stretch of 24 bases and resulted in the deletion of 6228 nucleotides encompassing part of IVS20, exons 21–22 and IVS22 (Figs. 4B and 4C). The BR963 proband was affected with breast cancer at age 40 and belonged to HBC family. Segregation analysis demonstrated that the mutation came from the maternal lineage (Fig. 5A). PCR amplification of genomic DNA from BR1154 patient resulted in an aberrant fragment of approximately 872 bp (also present in her mother, sample BR1148), whose direct sequencing confirmed loss of *BRCA1* exons 20, possibly originating from an erroneous homologous recombination process between an AluY (Alu family, SINE class; chr17:41205398-41205698) and an AluY (Alu family, SINE class; chr17:41205398-41205698) motifs. The rearrangement involved a perfectly repeated stretch of 11 bases and resulted in the deletion of 4173 nucleotides encompassing part of IVS19, exon 20 and IVS20 (Figs. 4D–4F). The BR1154 proband was affected with ovarian cancer at age 52 and belonged to a HBOC family (Fig. 5B). The segregation analysis demonstrated that the mutation originating from the maternal lineage segregated in three individuals (Fig. 5B). Finally, PCR amplification of genomic DNA from BR1379 patient, resulted in an aberrant fragment of approximately 2,027 bp, whose direct sequencing confirmed loss of *BRCA1* exons 16 and 17, possibly originated from an erroneous homologous recombination process between an AluSp (Alu family, SINE class; chr17:41224585-41224884) and an AluSg (Alu family, SINE class; chr17:41218424-41218724) motif. The rearrangement involved a perfectly repeated stretch of 16 bases and resulted in the deletion of 6155 nucleotides encompassing part of IVS15, exons 16-17 and IVS17 (Figs. 4G–4I). The BR1379 proband

Nicolussi et al. (2019), *PeerJ*, DOI 10.7717/peerj.7972

Peerj

**Table 2  CNVs prediction by IR software algorithm in TS and VS.** The confidence score is the probability that the number of copies of the region of interest is different from 2, which is the normal value, while the precision score indicates how much the algorithm is certain of the accuracy of the number of copies estimated by the analysis.

|     | Sample ID | Locus | Type | Genes | Location | Length | Copy number | CytoBand | CNV confidence | CNV precision |
|-----|-----------|-------|------|-------|----------|--------|-------------|----------|----------------|---------------|
|     | BR59 | chr17:41197602 | CNV | BRCA1 | exon 23-24 | 2.138 kb | 1 | 17q21.31 (41197602–41199740) × 1 | 5.66 | 5.66 |
| TS  | BR328 | chr17:41215277 | CNV | BRCA1 | exon 18-19 | 749 kb | 1 | 17q21.31 (41215277–41216026) × 1 | 13.05 | 13.05 |
|     | BR409 | chr17:41275973 | CNV | BRCA1 | exon 2 | 275 kb | 1 | 17q21.31 (41275973-41276248) × 1 | 1.14 | 1.14 |
| VS  | BR963 | chr17:41201074 | CNV | BRCA1 | exon 21-22 | 2.18 kb | 1 | 17q21.31 (41201074–41203254) × 1 | 9.14 | 9.14 |
|     | BR1379 | chr17:41215855 | CNV | BRCA1 | exon 16-18 | 7.44kb | 1 | 17q21.31 (41215855–41223295) × 1 | 5.11 | 5.11 |

**Table 3  Performance of NGS-dependent LGRs analysis.**

| | | Tot | MLPA | | Results |
|---|---|---|---|---|---|
| | | | LGR | No LGR | |
| DQ | LGR | 33 | 3 | 30 | 64.7% accuracy (95% CI [50.6–76.7]) |
| | | | | | 100% sensitivity (95% CI [22.8–98.4]) |
| | No LGR | 52 | 0 | 52 | 63.4% specificity (95% CI [49–75.8]) |
| | | | | | 100% NPV (95% CI [87.6–99.9]) |
| IR | LGR | 29 | 2 | 27 | 67.1% accuracy (95% CI [52.9–78.7]) |
| | | | | | 66.7% sensitivity (95% CI [8.9–98.8]) |
| | No LGR | 56 | 1 | 55 | 67.1% specificity (95% CI [52.7–78.9]) |
| | | | | | 98.2% NPV (95% CI [85.6–99.9]) |

**Notes.**

Validation metrics were defined as: Accuracy = (TP + TN)/(TP + FP + TN + FN); Sensitivity = TP/(TP + FN); Specificity = TN/(TN + FP); Negative Predictive Value = TN/(TN + FN), where TP, true positives; TN, true negatives; FP, false positives; FN, false negatives.

was affected with bilateral breast cancer at age 42 and 58 and belonged to a family with colon cancer and hepatomas cases (Fig. 5C).

In conclusion, our results in the VS allow us to propose an operative algorithm which uses DQ calculation and IR analysis to select samples to be subjected to MLPA analysis, as indicated in Fig. 6. Indeed, all DQ positive samples should be subjected to MLPA, while DQ and IR double positive samples, sharing calls in the same regions, could be directly subjected to second level confirmation assay or directly to breakpoint characterization. In principle, all DQ negative samples (52 sample out of 85 in our VS) could be considered negative for LGRs, thus completing the analysis at this step.

## DISCUSSION

A complete clinical level analysis of *BRCA1* and *BRCA2* in hereditary breast/ovarian cancer includes the study of LGRs. Many methods have been used to identify LGRs, such as fluorescent *in-situ* hybridization (FISH) and microarrays (*Xia et al., 2018*), Southern blot, long-range PCR, quantitative multiplex PCR of short fragments (QMPSF) (*Ewald et al., 2009*), semiquantitative multiplex PCR, real-time PCR, restriction analysis and sequencing (*Armour et al., 2002*). All these methods are limited by their low throughput, time consuming, large amounts of high molecular weight DNA request and several false negative results (*Ewald et al., 2009*). More recently a multiplex PCR-based method that allows the determination of copy number status of multiple loci in a single assay, has been developed by Multiplicom (http://www.multiplicom.com) and described as a valid method (*Concolino et al., 2014*). However, the MLPA represents the most widely used approach to scan for LGRs in *BRCA1*/2 genes (*Ruiz de Garibay et al., 2012*). The simultaneous detection of mutations and copy number alterations is an attractive and useful prospect for clinical settings. In the last years the NGS-based approaches for genetic testing offered a powerful alternative for *BRCA1/2* mutation detection. However, the specificity of this approach is still considered not completely satisfactory for a correct LGRs detection. One of the most relevant aspects concerns the library preparation method, with the amplicon-based

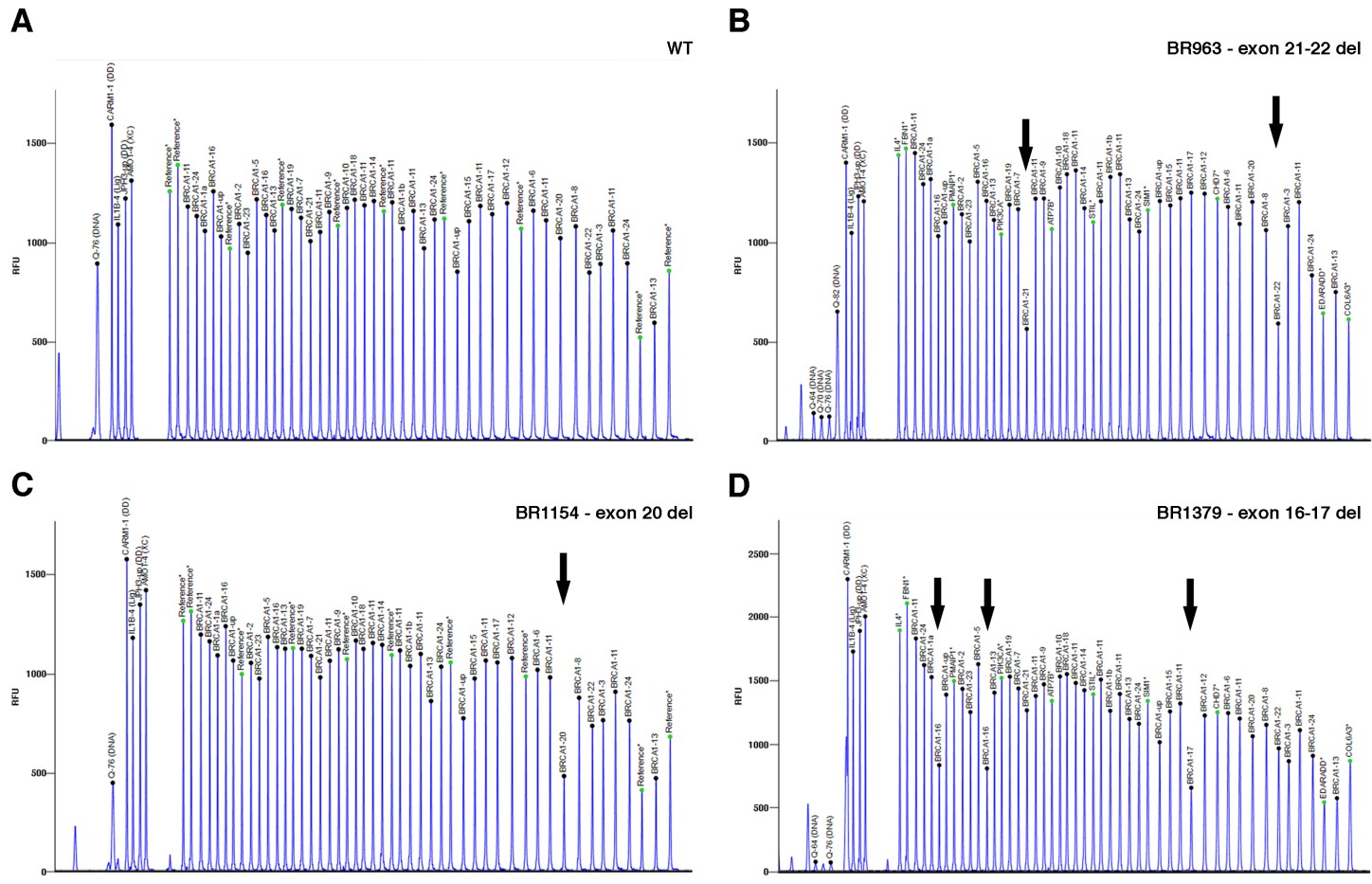

**Figure 2** **BRCA1 MLPA electropherogram showing aberrant profiles in BR963, BR1154, and BR1379 patients.** (A) Wild-type sample (WT). Black arrows indicate the deletion of (B) *BRCA1* exons 21-22 (BR963), (C) *BRCA1* exon 20 (BR1154), (D) *BRCA1* exons 16-17 (BR1379).

approach having a lower specificity compared to target enrichment approaches (*Apessos et al., 2018*). Here we reported the definition of an operative algorithm to use amplicon-based Ion-PGM/Ampliseq *BRCA1/BRCA2* sequencing data to efficiently predict the occurrence of *BRCA1* LGRs. By comparison of the results obtained with DQ and IR analyses, we demonstrate that DQ had 100% sensitivity and 100% NPV, at variance with IR analysis, which failed in the identification of a *BRCA1* exon 20 deletion. This result is consistent with one known limitation of the IR software, able to detect CNVs only if the region of interest is covered by more than one amplicon (https://assets.thermofisher.com/TFS-Assets/LSG/brochures/CNV-Detection-by-Ion.pdf). Indeed, *BRCA1* exon 20, deleted in BR1154 sample, is covered by only one amplicon in the Ion AmpliSeq™ BRCA1 and BRCA*2* Panel, making IR incapable of calling this CNV.

Of course, a major caveat deals with the limited specificity and accuracy of our approach, which could not overcome the limitations also reported by other groups (*Feliubadalo et al., 2013*; *Pilato et al., 2016*). Thus, although our operative algorithm cannot fully substitute for MLPA analysis, and if our data will be confirmed in larger data sets, we suggest that

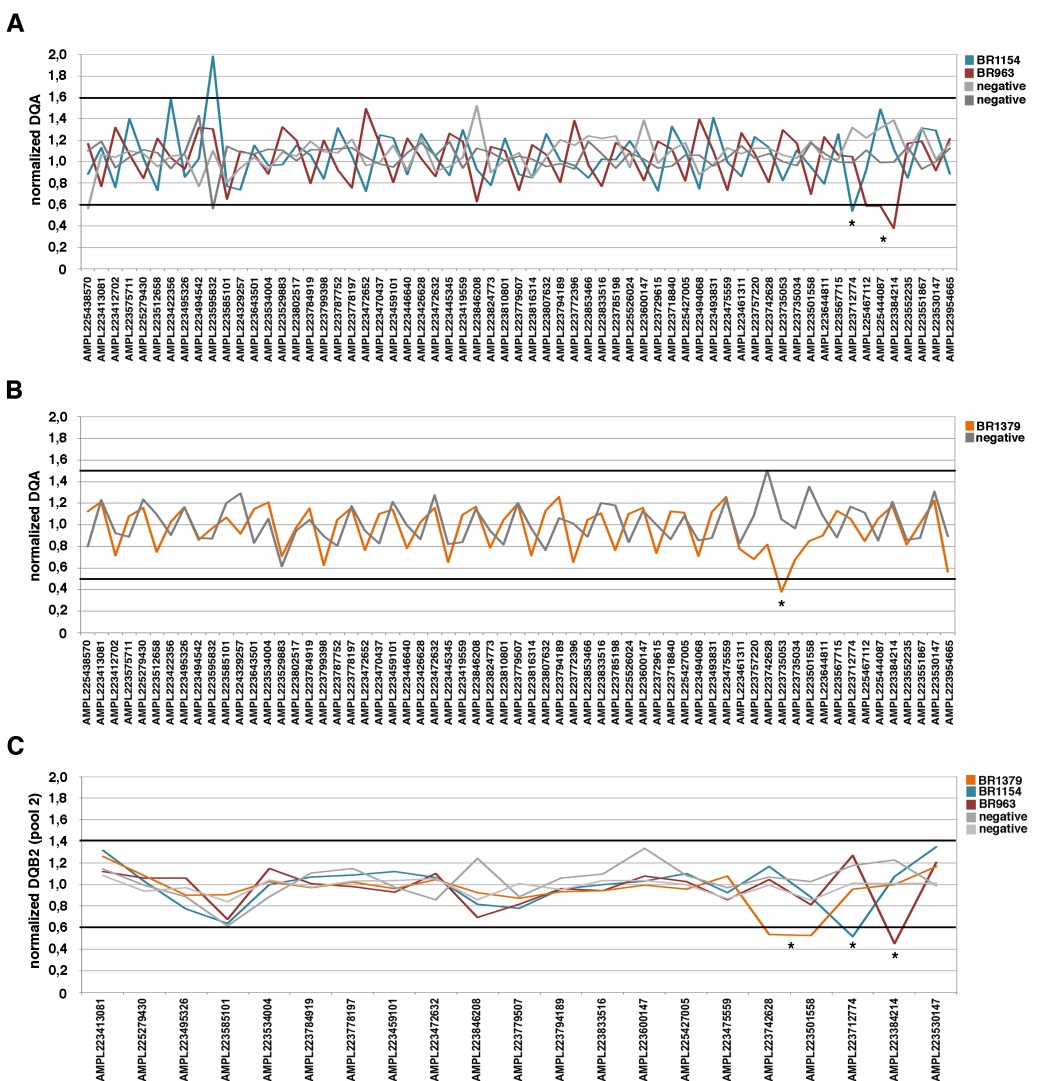

**Figure 3** **DQ analyses for the representative samples for VS.** (A) and (B) for each sample, every peak represents the ratio of the amplicon read count normalized on *BRCA1/BRCA2* total reads and the average of normalized amplicon read counts from all samples on a single chip (DQA). (C) for each sample, every peak represents the ratio of the amplicon read count normalized on *BRCA1/BRCA2* total reads and the average of the coverage data of a baseline built from 127 LGRs negative samples considering separately the amplicon pools (DQB2, pool 2). The threshold = mean ± 2 SD. Value > mean ± 2 SD is indicative of a duplication; Value < mean ± 2 SD is indicative of a deletion. * indicated the amplicons included in the region involved in the rearrangement as confirmed by MLPA analysis.

combined DQ and IR analyses could be used for selecting samples to be subjected to MLPA analysis following the flow chart depicted in Fig. 4, with significant savings in time and money.

Another important contribution of this paper is the molecular characterization of the three novel *BRCA1* rearrangements up to providing their unique breakpoint coordinates. Deletion of exons 21 and 22 causing damage to the C-terminal BRCT domain of the

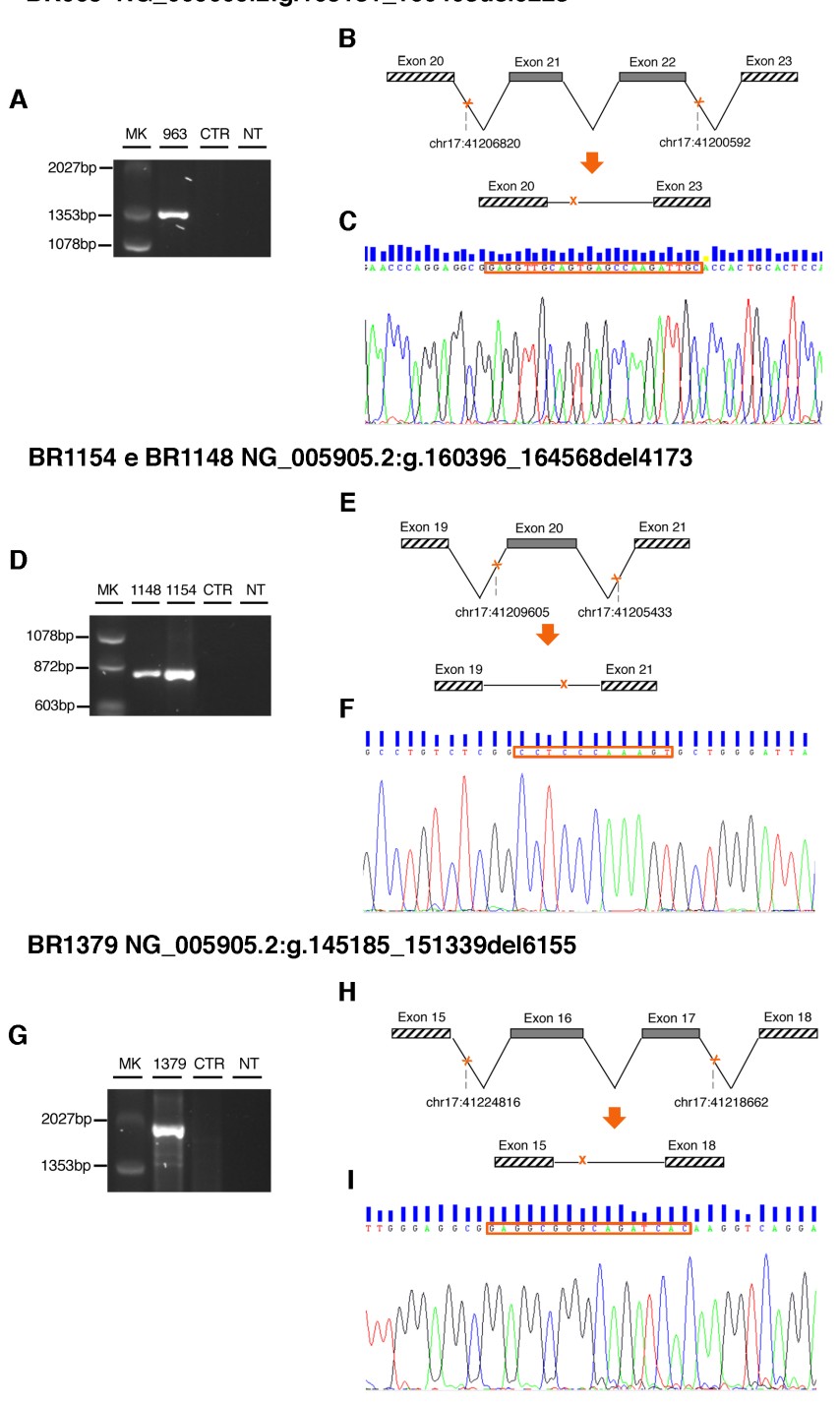

**BR963  NG_005905.2:g.163181_169408del6228**

A

**BR1154 e BR1148 NG_005905.2:g.160396_164568del4173**

D

**BR1379 NG_005905.2:g.145185_151339del6155**

G

**Figure 4  Characterization of BRCA1 LGRs.** (A) Gel image of PCR products. PCR amplification of the genomic region spanning the *BRCA1* rearrangement resulted in a fragment of approximately 1,353 bp present only in the proband BR963. (B) and (C) schematic representation and electropherogram showing the *BRCA1* exons 21 and 22 deletion. 

**Figure 4 (...continued)**
The variant arose from an erroneous homologous recombination process between an AluSq2 (Alu family, SINE class; chr17:41206762-41207066) and an AluSz (Alu family, SINE class; chr17:41200521-41200834) motif, and it involved a perfectly repeated stretch of 24 bp. (D) Gel image of PCR products. PCR amplification of the genomic region spanning the *BRCA1* rearrangement resulted in a fragment of approximately 872 bp present in the proband BR1154 and in her mother BR1148. (E) and (F) schematic representation and electropherogram showing the *BRCA1* exon 20 deletion. The variant arose from an erroneous homologous recombination process between two AluY motif at chr17:41205398-41205698 and chr17:41205398-41205698, respectively, and it involved a perfectly repeated stretch of 11 bp. (G) Gel image of PCR products. PCR amplification of the genomic region spanning the *BRCA1* rearrangement resulted in a fragment of approximately 2027 bp present only in the proband BR1379. (H) and (I) schematic representation and electropherogram showing the *BRCA1* exons 16 and 17 deletion. The variant arose from an erroneous homologous recombination process between an AluSp motif (Alu family, SINE class; chr17:41224585-41224884) and an AluSg (Alu family, SINE class; chr17:41218424-41218724) motif, and it involved a perfectly repeated stretch of 16 bp. MK, marker; NT, no template; CTR healthy individual DNA.

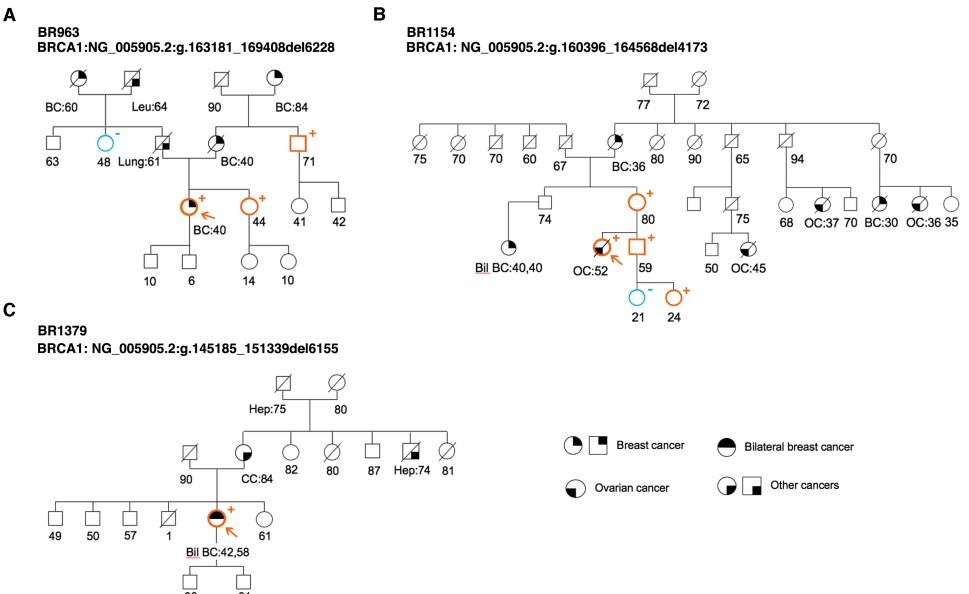

**Figure 5** **Pedigree of the HBC or HBOC family carriers of *BRCA1* novel LGRs.** (A) Exons 21–22 deletion (BR963). (B) Exon 20 deletion (BR1154). (C) Exons 16–17 deletion (BR1379). Probands are indicated with an arrow. Cancer type and age at diagnosis are reported and described as: BC, breast cancer; Pan, pancreas; Leu, leukemia; Lung; bil BC, bilateral breast cancer; OC, ovarian cancer; Hep, hepatoma; CC, colon cancer.

BRCA1 protein has been reported and characterized in Czech (*Vasickova et al., 2007*; *Ticha et al., 2010*) and Malay population (*Hasmad et al., 2015*), but with different breakpoints. *BRCA1* exon 20 deletion has been described in Italian and Greek population (*Montagna et al., 2003*; *Belogianni et al., 2004*; *Armaou et al., 2007*) but all different from each other and from our own, with respect to their breakpoints. The B*RCA1* exons 16-17 deletion, responsible of BRCA1 loss of function (*Carvalho et al., 2009*), has been reported in Latin America/Caribbean population, but the breakpoints were not provided by the authors (*Judkins et al., 2012*). Similar to many other cases (*Mazoyer, 2005*; *Buffone et al., 2007*;

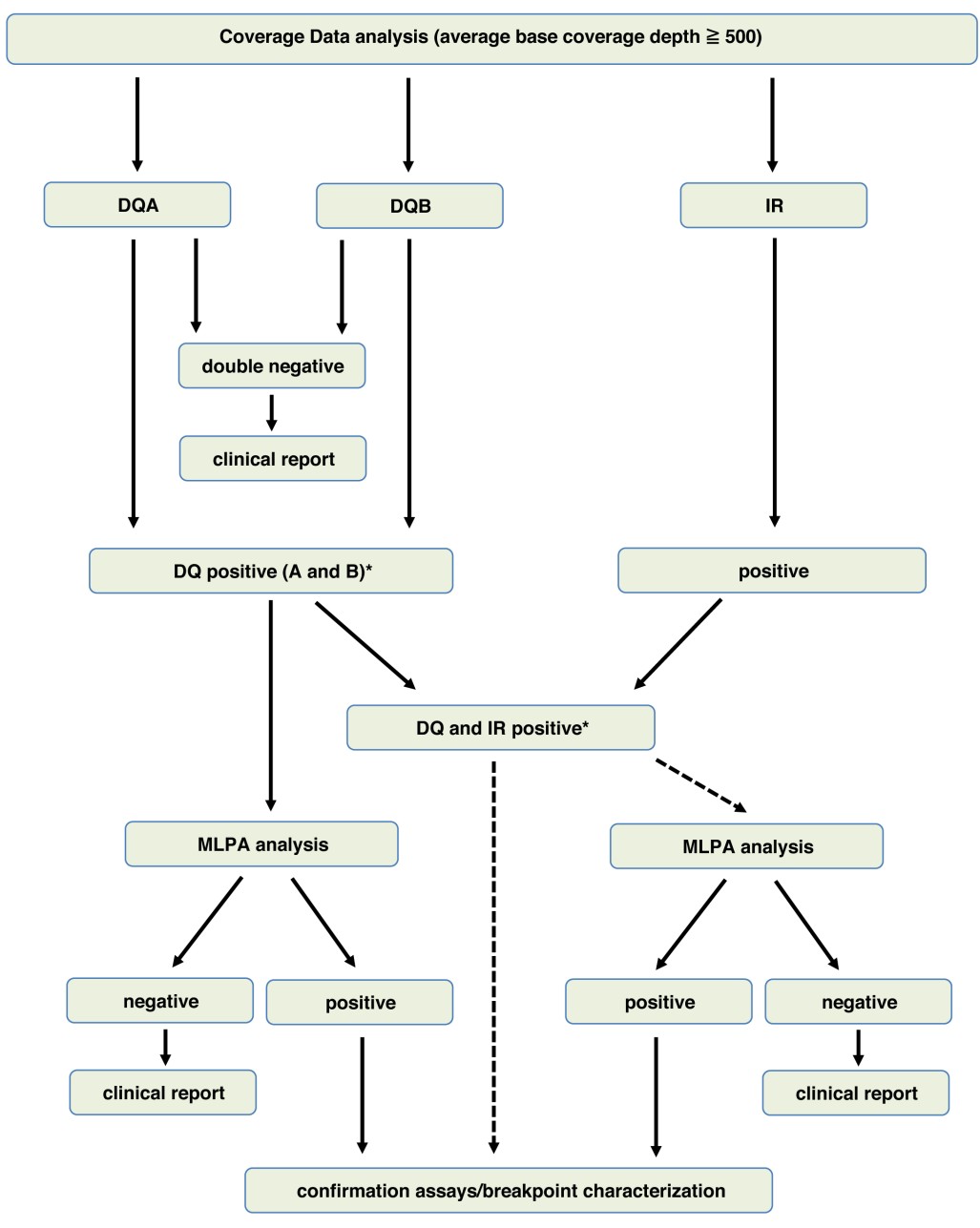

**Figure 6** Operative algorithm to select samples for MLPA analysis.

*Ewald et al., 2009*), all three novel rearrangements described here, are likely to be due to an erroneous homologous recombination event between perfectly matching Alu repeats.

## CONCLUSION

In conclusion, here we described a simple approach that require the use of a basic statistical package such as Microsoft Excel, to predict the occurrence of LGRs by the analysis of NGS data designed for Ion AmpliSeq™ BRCA1 and BRCA2 Panel/IT-PGM platform, applicable

to all NGS platforms in use to reduce the number of samples to be subjected to MLPA analysis. We also characterized for the first time the breakpoints of three novel *BRCA1* LGRs.

## ACKNOWLEDGEMENTS

The authors would like to thank Dr. Alessandro Albiero for advice on data analysis.

### Funding

This work was supported by the Italian Ministry of Education, Universities and Research—Dipartimenti di Eccellenza—L. 232/2016; Associazione Italiana per la Ricerca sul Cancro (AIRC) grant IG17734, Italian Ministry of University and Research, PRIN projects, and Istituto Pasteur-Fondazione Cenci Bolognetti (to Giuseppe Giannini); Francesca Fabretti is the recipient of a fellowship of the PhD Programme in Tecnologie Biomediche in Medicina Clinica, University La Sapienza. The funders had no role in study design, data collection and analysis, decision to publish, or preparation of the manuscript.

### Grant Disclosures

The following grant information was disclosed by the authors:
Italian Ministry of Education, Universities and Research—Dipartimenti di Eccellenza: L. 232/2016.
Associazione Italiana per la Ricerca sul Cancro (AIRC): IG17734.
Italian Ministry of University and Research.
PRIN projects.
Istituto Pasteur-Fondazione Cenci Bolognetti (to Giuseppe Giannini).
Tecnologie Biomediche in Medicina Clinica, University La Sapienza.

### Competing Interests

The authors declare there are no competing interests.

### Author Contributions

- Arianna Nicolussi conceived and designed the experiments, performed the experiments, analyzed the data, prepared figures and/or tables, authored or reviewed drafts of the paper, approved the final draft.
- Francesca Belardinilli, Yasaman Mahdavian, Virginia Valentini and Sonia D'Inzeo performed the experiments, prepared figures and/or tables, approved the final draft.
- Valentina Silvestri analyzed the data, contributed reagents/materials/analysis tools, authored or reviewed drafts of the paper, approved the final draft.
- Marialaura Petroni, Massimo Zani, Sergio Ferraro, Stefano Di Giulio and Francesca Fabretti performed the experiments, contributed reagents/materials/analysis tools, prepared figures and/or tables, approved the final draft.
- Beatrice Fratini, Angela Gradilone analyzed the data, contributed reagents/materials/-analysis tools, prepared figures and/or tables, approved the final draft.

- Laura Ottini, Giuseppe Giannini conceived and designed the experiments, authored or reviewed drafts of the paper, approved the final draft.
- Anna Coppa and Carlo Capalbo conceived and designed the experiments, analyzed the data, authored or reviewed drafts of the paper, approved the final draft.

### Human Ethics

The following information was supplied relating to ethical approvals (i.e., approving body and any reference numbers):

The Ethics Committee of the University of Roma "La Sapienza" approved this research (Prot.: 88/18; RIF.CE:4903, 31-01-2018).

### DNA Deposition

The following information was supplied regarding the deposition of DNA sequences:

The following LGRs are available at ClinVar:

NG_005905.2:g.163181_169408del6228: ClinVar; Variation ID 598936; NG_005905.2:g.145185_151339del6155: ClinVar; Variation ID 598935; NG_005905.2:g.160396_164568del4173: ClinVar; Variation ID 598937

### Data Availability

The raw measurements are available in the Supplemental Files.

### Supplemental Information

Supplemental information for this article can be found online at http://dx.doi.org/10.7717/peerj.7972#supplemental-information.

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
