# Peer review of "Identification of novel BRCA1 large genomic rearrangements by a computational algorithm of amplicon-based Next-Generation Sequencing data"

_PeerJ, doi:10.7717/peerj.7972_

## Round 0.1 · original submission · Minor Revisions

It would add greatly to your paper if you could analyse an additional series of known BRCA1 CNVs found by MLPA.

Reviewer 1 ·

Basic reporting

Nicolussi A. and Colleagues present the manuscript entitled “Identification of novel BRCA1 large genomic rearrangements by a computational algorithm of amplicon based Next-Generation Sequencing data” where they reported a computational algorithm to predict the presence of BRCA1 LGRs using NGS data. The paper is interesting and well organized, the figures and tables are clear, literature references sufficient and the English language is of a good standard. Therefore, it requires minor changes.


MINOR COMMENTS:

1. Line 43: add a space between exons and 16;
2. Line 200: change (Fig.1A) in (Fig.1a);
3. Legend Figure 3: write c in bold.

Experimental design

No comment

Validity of the findings

No comment

Additional comments

The reserch is original and well defined. The data are robust and the statistical analysis is reliable. Methods are described with sufficient details and informations. However your discussion needs more detail. I suggest that you improve the discussion providing an overview and a comment on the current technologies to detect BRCA1/2 LGRs.

Reviewer 2 ·

Basic reporting

I have read the paper by Nicolussi et al. " Identification of Novel BRCA1 Large genomic rearrangements by a computational algorithm of Amplicon-Based Next-Generation Sequencing data”. This study is interesting and focuses on a topic that currently plays a very important growing role in biomedical research and diagnostics. The paper is scientifically accurate, complete and fully comprehensive to the reader. The reference section is adequate, updated and appropriate to back up the points made in the article.

Experimental design

The methodological approaches and techniques used in this study are appropriate. The authors described a DQ-based algorithm able to identify BRCA1 LGRs, translating NGS clinical sequencing via an Ion Torrent PGM platform into a suitable tool. By this approach, applied to a consecutive cohort of 85 uncharacterized probands (validation set), they predicted the presence of BRCA1 LGRs with 100% sensitivity and 100% negative predictive value, identifying 3 novel BRCA1 LGRGs. Moreover, the authors characterized the breakpoints of these 3 novel BRCA1 deletions, involving exons16-17, 21-22 and 20, respectively.

Validity of the findings

Results are original and provide useful information to the field. The authors address an important topic. Given the increasing demand for genetic diagnostics, an appropriate and complete diagnostic screening of BRCA1/2 germline mutations ensuring quality, is a pressing issue.

Additional comments

I have no major comments on this paper. However, I provide several suggestions in order to make the paper more attractive to readers:
• The algorithm depicted in Figure 4 includes the breakpoints characterization, therefore it may be better to move Figure 4 and its description at the end of the last paragraph of Results.
• Could the authors check the use of abbreviations: sometimes they are missed.
• A minor detail is the varying use of the terms CNV and LGRs in Abstract, Figures and Tables, which may be confusing.
• Could the authors make a comment on whether they feel this approach can be applied to DNA extracted from cancer/tumor tissues?

---

## Round 0.2 · accepted · Accept

You have satisfactorily addressed the reviewer comments. The gene symbols in the added text need italicizing though '..BRCA1/2 genes (Ruiz de Garibay et al., 2012)'